# Comprehensive Assessment of Water Quality and Pollution Source Apportionment in Wuliangsuhai Lake, Inner Mongolia, China

**DOI:** 10.3390/ijerph17145054

**Published:** 2020-07-14

**Authors:** Rui Shi, Jixin Zhao, Wei Shi, Shuai Song, Chenchen Wang

**Affiliations:** 1State Key Laboratory of Urban and Regional Ecology, Research Center for Eco-Environmental Sciences, Chinese Academy of Sciences, Beijing 100085, China; s18247808609@163.com (R.S.); wccapp@126.com (C.W.); 2Environmental Information Monitoring Center of Bayannur, Bayannur 015000, China; 3Institute of Environmental Science of Bayannur, Bayannur 015000, China; zhaojixin1984@163.com; 4Institute of Loess Plateau, Shanxi University, Taiyuan 030006, China; shiwei@sxu.edu.cn; 5University of Chinese Academy of Sciences, Beijing 100049, China

**Keywords:** water quality, principal component analysis, cluster analysis, comprehensive assessment, Wuliangsuhai Lake

## Abstract

Water quality is a key indicator of human health. Wuliangsuhai Lake plays an important role in maintaining the ecological balance of the region, protecting the local species diversity and maintaining agricultural development. However, it is also facing a greater risk of water quality deterioration. The 24 water quality factors that this study focused on were analyzed in water samples collected during the irrigation period and non-irrigation period from 19 different sites in Wuliangsuhai Lake, Inner Mongolia, China. Principal component analysis (PCA) and hierarchical cluster analysis (HCA) were conducted to evaluate complex water quality data and to explore the sources of pollution. The results showed that, during the irrigation period, sites in the middle part of the lake (clusters 1 and 3) had higher pollution levels due to receiving most of the agricultural and some industrial wastewater from the Hetao irrigation area. During the non-irrigation period, the distribution of the comprehensive pollution index was the opposite of that seen during the irrigation period, and the degree of pollutant index was reduced significantly. Thus, run-off from the Hetao irrigation area is likely to be the main source of pollution.

## 1. Introduction

Water quality is affected by both anthropogenic activities and natural factors, with the latter influencing surface water quality through industrial sewage, pesticides, chemical fertilizers, and the increased exploitation of water resources [1,2,3]. Such influences have led to a decrease in water quality, generating great pressure on the structure and function of aquatic ecosystems [4,5,6]. Therefore, the implementation of regular water monitoring programs and reasonable assessment of chemicals could help control water pollution and restore aquatic ecosystems.

Conventional quality classes are comprised of water quality regulations with set limits between them, although these are inherently imprecise. Besides, not all water quality factors could be included in a single class [7], which can lead to confusion in defining the quality of sampling sites. The application of multivariate statistical techniques, such as hierarchical cluster analysis (HCA) and principal component analysis (PCA), has helped identify possible sources that influence water systems, and have offered valuable tools for use in the reliable management of water resources [8,9,10]. Many studies showed these methods could be effectively used to evaluate water quality factors and to explore similarities among different samples [11,12,13].

Wuliangsuhai Lake (108°43′–108°57′ E, 40°36′–41°03′ N) is large freshwater lake [14] (Figure 1a), with a mean water depth of less than 2 m and a surface area of 293 km^2^. It plays an important role in maintaining the ecological balance of the region, protecting the local species diversity and maintaining agricultural development. Wuliangsuhai Lake is important in terms of the aquatic plants, fisheries, and birds that it provides habitats for, as well as the tourism resources that have developed around it. However, it also has an important role as an ecological barrier in northern China and is a supplemental water source for the Yellow River in Inner Mongolia during the dry season [15]. The lake is an important part of the Hetao Irrigation area, into which >90% of the surrounding farmland is drained. Such agricultural discharges then drain into the Yellow River; therefore, the water pollution assessment, source analysis and water quality improvement in Wuliangsuhai have received extensive attention, including N, P, chemical oxygen demand (COD) [16], microplastics [17], surfactants, and heavy metals [18,19]. Previous research [20] has shown that the absence of effective drainage and uncontrolled irrigation resulted in a rising groundwater table, the expansion of the lake, and severe non-point-source pollution. Some researchers also evaluated the seasonal changes of nutritional status and water quality factors [21,22] and found that the eutrophication status in spring was significantly higher than that in other seasons, which was mainly related to transparency and COD. Wu et al. [23,24] simulated monthly streamflow, nitrogen, and phosphorus in Wuliangsuhai watershed to evaluate the effects of fertilization management on pollutant reduction. Besides, the effects of microorganisms, adsorption–desorption dynamics, and irrigation strategies on the water quality factors have been of concern in this region [25,26,27,28]. However, there are still large unknowns regarding water quality characteristics and their influencing factors among the irrigation and non-irrigation periods. Therefore, the comparative studies before and after irrigation could be of great help to understand the non-point source processes and pollutant reduction of the Wuliangsuhai watershed.

In this study, we took Wuliangsuhai Lake as an example using multivariate statistical techniques to assess the water quality factors. The objectives of this study are: (1) to analyze the changes of 24 hydrochemical variables, as well as to evaluate variables about the similarities and dissimilarities among various water quality factors; (2) to identify parameters specific to analyzing the spatio-temporal dissimilarity in Wuliangsuhai Lake; and (3) to explore the influence of pollution sources on the water quality parameters, and assessed water quality using the Nemerow pollution exponential method.

## 2. Materials and Methods

### 2.1. Sample Collection and Chemical Analytical Procedures

Water samples were collected from 19 sites (Figure 1b) at bimonthly intervals between June 2015 and November 2015 in Wuliangsuhai Lake. Each sample was composed of three mixed water columns at depths of 0.5 m, 1 m, and 2 m. The samples were kept in polyethylene plastic bottles that had been previously cleaned with metal-free soap, rinsed repeatedly with distilled water, soaked in 10% nitric acid for 24 h, and finally rinsed with ultrapure water. All water samples were maintained in a refrigerator at 0 °C during transportation to the laboratory, and then later for processing and analysis. We measured the temperature, transparency, pH, turbidity, and dissolved oxygen (DO) of water samples using field instruments, including an electronic thermometer, transparency meter, digital pH meter, turbidity meter, and DO meter, respectively. Other physical and chemical parameters analyses were carried out according to the China Nation Surface Water and Wastewater Monitoring technology standards (HJ/T91-2002) and the National Surface Water Environment Quality Standards (GB3838-2002) (Table 1). Reagent/procedural blanks and three control samples were analyzed for anionic surfactant, suspended matter, cyanide, total nitrogen (TN), total phosphorus (TP), KMnO_4_, petroleum, volatile phenol, sulfide, fluoride, COD, and other trace elements. All water samples were analyzed within 24 h of collection.

A total of 28 hydrochemical variables were analyzed; however, the concentrations of volatile phenols, cyanide, sulfide, and Cr^6+^ for all samples were below detection levels (0.001 mg/L for volatile phenol and cyanide, 0.02 mg/L for sulfide, and 0.004 mg/L for Cr^6+^). Thus, only 24 variables were analyzed further.

### 2.2. Multivariate Statistical Analyses

The statistical analysis and mathematical computations of water quality factors were conducted using SPSS 22.0 (SPSS Inc., Chicago, IL, USA). Multivariate analysis of the data set was performed using PCA and HCA, and nonparametric tests (Kruskal–Wallis H method [29]) were used when the data collected did not meet the requirements of a normal distribution and homogeneity of variance. The Conover–Iman test was performed as a post-hoc test to reveal which pairs of indicators were significantly different. PCA was applied for the assessment of the irrigation effects on water quality, and each autoscaling parameter was calculated using Equation (1) before PCA to minimize the influence of different variables and their respective units of measurements. HCA was conducted to analyze spatial similarity based on the PCA result for grouped sampling sites. The results from the PCA were used to understand the differences in water quality between the different sample periods.
(1)$$z=\frac{C-\mathrm{Mean}}{\sigma }$$
where C represents the water quality factors; Mean is the mean value of C; and σ is the standard deviation of *C*.

#### 2.2.1. PCA

PCA is a method of mathematical transformation that attempts to reduce the dimensionality of datasets. The set of relevant variables is converted into another equal number of independent variables that are not related by a linear transformation, and these new variables are then arranged in descending order of variance [30]. Each variable is held constant in the mathematical transformation of the total variance; the first variable that has the greatest variance is designated the first principal component, the second largest variance of the second variable that is not related to the first variable is the second principal component, and so on. PCA aims to determine the fewest variables that explain the majority of the variance in the original data.

#### 2.2.2. HCA

HCA is the process of grouping a set of physical or abstract objects into classes of similar objects. It could divide a large number of samples into reasonable classifications according to their respective characteristics, and the objects in the same cluster have great similarity [31,32]. The dendrogram provides a summary of clustering processes, presenting a visual picture of the groups and their proximity. In this study, HCA was carried out on the normalized values through Ward’s method, using squared Euclidean distances as a measure of dissimilarity.

### 2.3. Improved Nemerow Pollution Exponential Method and Comprehensive Evaluation

The Nemerow pollution exponential method is one of the most commonly used comprehensive assessment approaches, which allows the assessment of the overall degree of water pollution and includes the contents of all analyzed water quality factors [33,34]. In traditional research, the Nemerow comprehensive pollution index is calculated by the concentrations of pollutants and the standard values, and then obtains the weighted averages of these indexes to get the total pollution index. This method highlights the influence of the largest pollution factor on the results and does not consider the disadvantage of weight factors. In this study, the factor with the largest weight (F_w_ max) among all water quality factors was determined according to Equations (2) and (3). The *C_i_/S_i_* ratio of the F_w_ was considered in Equation (4). Then, the improved Nemerow comprehensive pollution index was calculated as shown in Equations (4) and (5). The average values determined for *P*_total_ were conducted to assess the water quality of the lake (Table 2).
(2)$$\omega_{i}=r_{i}/\sum_{i=1}^{m}r_{i}$$
(3)$$r_{i}=S_{max}/S_{i}$$
(4)$$P_{i}=\sqrt{\left[{F_{i}}^{2}+{\left(\frac{F_{max}+F_{w}}{2}\right)}^{2}\right]/2}$$
(5)$$F_{i}=\frac{C_{i}}{S_{i}}\;\;\;\;\;\;\;\;\;i=1,\;2,\ldots ,\;n$$
where *F_i_* represents the single factor pollution index of element *i; C_i_* is the measured water quality factors (mg/L, pH: dimensionless, coliform bacteria: Ind); *S_i_* represents the guideline value (Nation Surface Water Environmental Quality standard of China) of element *i*; *S_max_* represents the highest guideline value; *P_i_* represents the Nemerow pollution index; *F_max_* represents the maximum value of the single factor pollution index; *F_w_* represents the single factor pollution index with the largest weight among all factors. ω_i_ is the weight of the water quality factor, *r*_i_ is the correlation ratio of the water quality factor, and *n* is the quantity of the factors.
(6)$$P_{total}={\frac{1}{n}}_{i}\overset{n}{\underset{i=1}{\sum }}P_{i}$$
where *i* is the number of sampling site, and *P*_total_ is the comprehensive pollution index of the lake [32,33,35].

## 3. Results and Discussion

### 3.1. Descriptive Statistics of Water Quality Factors in Wuliangsuhai Lake

The basic statistics calculated for the water quality of Wuliangsuhai Lake are summarized in Table 3. The pH values of the collected water samples ranged from 7.89 to 9.31, exceeding the limit range of 6–9 allowed by the National Surface Water Environment Quality Standards for water quality. The suspended matter, salinity, and transparency during the sampling period showed the greatest range, with values of 97 mg/L, 5128 mg/L, and 180 mg/L, respectively. The concentrations of heavy metals (Cu, Zn, Cr^6+^, and Se), As, and anionic surfactants in most samples were below the detection limit, whereas the maximum concentrations of Hg and Pb were close to the permissible limit of State Environmental Protection Administration (SEPA). The average concentrations of nutrients (TN, TP, NH_3_–N, and chlorophyll a) were higher than the guide levels, with the maximum over standard multiples being 5.54, 1.11, 1.67, and 12, respectively. The concentration levels of COD_mn_, KMnO_4_, and BOD are of important concern, because these indicators represent the degree of the biological, chemical and physical pollution in the lake. However, the maximum values of these factors were 5.65, 2.05, and 2.87, respectively, which exceed the national water quality standards. Therefore, the study area has been contaminated with relatively high pollution levels at some sites. COD_mn_ refers to the amount of oxidant consumed in the treatment of water samples with the strong oxidant, which was used to indicate the degree of mixed pollution. The high level of BOD is due to the development of local fisheries and the discharge of waste from living sources.

The coefficients of variation for NH_3_–N, oil, anionic surfactants, Se, Zn, coliform bacteria, Pb, Cu, chlorophyll, Cd, and total suspended solids (TSS) values were relatively high, indicating these water quality factors were significantly different in terms of their temporal and spatial distribution. Thus, changes in point and non-point source pollution in the watershed may cause the water quality of Wuliangsuhai to deteriorate significantly throughout the study.

The nonparametric test showed that significant differences (*p* < 0.05) were found in parameters including pH, turbidity, salinity, transparency, chlorophyll a, DO, KMnO_4_, BOD, COD_Mn_, TP, Fluoride, As, Hg, Pb, Cu, Zn, Cd, and Se among different sampling periods (Appendix A), indicating that there was a significant difference in the temporal distribution of these variables. When the variables were grouped in terms of the sampling sites, variables with a significant effect (*p* < 0.05) on water quality were pH, turbidity, TSS, salinity, transparency, KMnO_4_, BOD, COD_Mn_, TN, TP, and fluoride. Among these eleven parameters, the Conover–Iman post-hoc test showed significant pairwise differences (*p* < 0.05) between all variables. These results may indicate that sufficient variability of the data was captured and show the significant spatial differences of these variables.

### 3.2. Principal Component Analysis (PCA) for Irrigation and Non-Irrigation Periods

The PCA results based on the correlation matrix in the irrigation period and non-irrigation period are shown in Figure 2 and Appendix A. The first three components of the PCA analysis explained 36.945% of the total variance. Each component was characterized by the water quality parameters with higher-weighted values (Appendix A). Thus, the first three principle components were dominated by BOD, COD_Mn_, and salinity (PC1); turbidity, TSS, transparency, and TP (PC2); and pH, Zn, and Cd (PC3). Different contributions of PCA components during non-irrigation and irrigation periods were observed in Figure 2. The measured data of irrigation periods were characterized by PC3, while those of non-irrigation were characterized by PC2 and PC1. Heavy metal parameters (Zn and Cd), pH, and nutrient parameters reflected the degree of pollution during the irrigation period (Appendix A), suggesting that eutrophication is a significant problem during this stage. This is likely to be because the Hetao agriculture irrigation area uses a large amount of nitrogen fertilizer, explaining the high weighting of TN during the irrigation period. Numerous studies have demonstrated that there is an obvious positive correlation between TN and degree of eutrophication [21,22,36], with a utilization rate of nitrogen fertilizer being less than 40%; thus, a large amount of nitrogen in farmland sewage flows into Wuliangsuhai Lake, resulting in its eutrophication. COD pollutants comprise daily life excreta (67.2%; 13,481.54 tones) and industry excreta (32.8%; 6572.86 tones) [37]. Given that heavy metals were weighted more heavily during the non-irrigation period, we can conclude that these heavy metals flow into Wuliangsuhai Lake alongside sewage [38]. TN and NH_3_–N were both highlighted by PCA, and both are carried as sewage into the lake during the irrigation period and are released into the sediment during the non-irrigation period. In general, the water parameters highlighted by PCA reflect the degree of pollution during the non-irrigation period and irrigation period, with heavy metals and eutrophication factors mainly resulting from the discharge of agriculture sewage.

### 3.3. Hierarchical Cluster Analysis (HCA) for Irrigation and Non-Irrigation Periods

All sampling sites were divided into four groups for both the irrigation period and non-irrigation period (Figure 3), which constructed two dendrograms (Appendix A) using Ward’s method. In the irrigation period (Figure 3b), Cluster 1 corresponded to sites 1, 8, 9, and 16; Cluster 2 included sites 4, 14, 15, and 17, mainly located in the north and south of the lake; Cluster 4 contained sites 3, 5, 6, 7, 10, 11, 13, 18, and 19, which were located at the center and outlet of the lake. In the non-irrigation period (Figure 3b), Cluster 1 corresponded to sites 1, 2, 3, and 4; Cluster 2 contained sites 5 and 6, which were located at the north of the lake. Cluster 3 and Cluster 4 were located at the center and south parts of the lake. Compared with irrigation and non-irrigation periods, the spatial variations of the four groups were relatively large, which implied the potential impacts of non-point source pollution caused by flood irrigation on the water quality grouping of the lake. In the non-irrigation period, the water mainly comes from the north inlet, so the clusters are distributed regularly from south to north.

However, during the non-irrigation period, the lake receives farmland drainage mainly from three inlets that are close to sampling sites 1, 3, and 12. Therefore, HCA revealed that the water quality analyzed at these sites was affected by different pollutant sources in the irrigation and non-irrigation periods.

### 3.4. Possible Sources of Pollutants and Comprehensive Evaluation of Wuliangsuhai Lake

The sampling sites were divided by location during the different water periods, with a certain differentiation seen among the sites (Figure 3), which demonstrated farmland drainage is the main source of pollutants in Wuliangsuhai Lake, which also determines the temporal and spatial distribution of pollutants. This conclusion of the source analysis is similar to other studies [23,35]. The water quality of the sampling site located in the center of the lake was likely influenced by water inflow from the inlet of the Hetao irrigation area and by the environment of the lake sediment [18,25], where three farmland drainage channels converge. The water quality at the edge of the lake near the village was likely to have resulted from effluents from nonpoint sources. Water quality in sites 12 and 18 was also influenced by the many tourist attractions around these locations. The comprehensive pollution indices of sites 1, 2, 3, and 5 indicated that these sites were only slightly polluted (Figure 4, Appendix A). During the non-irrigation period, the comprehensive pollution index results showed a similar pattern during the irrigation period, although the degree of pollution was reduced (Figure 4). Therefore, it can be concluded that the water inflow from the Hetao irrigation sector is the main pollutant source for Wuliangsuhai Lake.

The comprehensive pollution levels of Clusters 2 and 3 were lower, whereas Clusters 1 and 4 were higher during the irrigation period, and some sampling sites were seriously polluted. Accordingly, comprehensive pollution values were better around the edges and in the northern section of the lake than in the center or southern areas. This implies that to reduce the number of sampling points and the related cost of the monitoring network operation, only one or two stations are needed in each group to estimate the water quality of the whole lake. HCA could provide a reliable classification of the surface water sampling points in the study area.

## 4. Conclusions

Most of the previous studies on Wuliangsuhai Lake were focused on a small number of major pollutants, such as COD, N, P, and relatively few sampling sites. Few studies have been concerned with the changes of a relatively large number of water quality parameters among the irrigation and non-irrigation periods. In this study, 24 hydrochemical variables, including heavy metals, N, P, and COD, were obtained from 19 sampling sites between irrigation and non-irrigation periods in Wuliangsuhai Lake. The spatial and temporal distribution of water quality in Wuliangsuhai Lake was studied based on multivariate statistical methods. The HCA method was used to divide 19 sampling sites into four groups, which could provide an optimal sampling strategy for the future, thereby reducing the number of samples and related costs. The principal component analysis helped identify the factors or sources of concentration changes of pollutants. We concluded that the main sources of pollution were likely to be industrial and agricultural sewage from the irrigation area and the local villages around the lake through a field investigation and data analysis. The study demonstrated that statistical analysis methods were useful in interpreting complex data sets, identifying pollution sources, and understanding changes in water quality. The results also showed that measures to reduce the anthropogenic emissions of pollutants should be taken. Otherwise, high pollution levels have the potential to affect high-quality socio-economic development. In the next step, the relationship between pollution sources around the lake and the spatial-temporal distribution of pollutant concentrations in the lake should be given more attention. Meanwhile, the characteristics of pollutant transports and chemical morphological changes in the Wuliangsuhai basin should be clearly explained for providing references for future planning and management.

## Figures and Tables

**Figure 1 ijerph-17-05054-f001:**
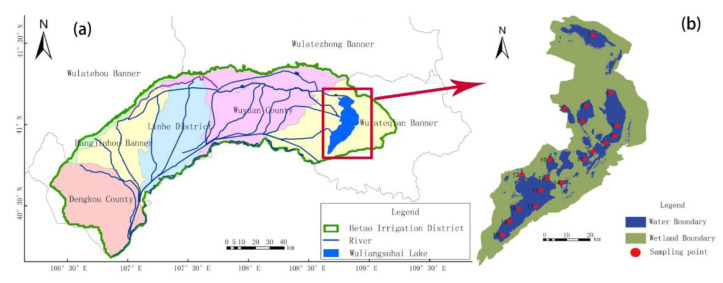
(**a**) The location of Wuliangsuhai Lake. (**b**) The sampling sites in Wuliangsuhai Lake.

**Figure 2 ijerph-17-05054-f002:**
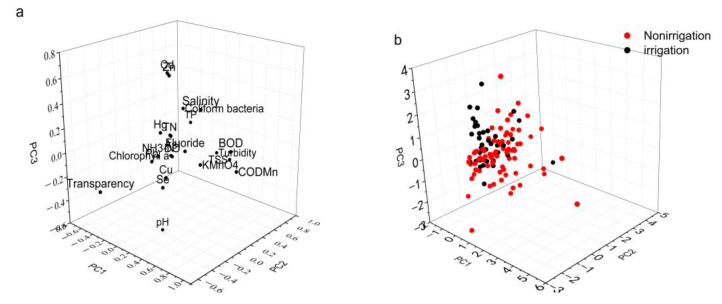
Factor score and loading plot of Principal Component Analysis (PCA) for water quality factors (**a**) and irrigation periods (**b**).

**Figure 3 ijerph-17-05054-f003:**
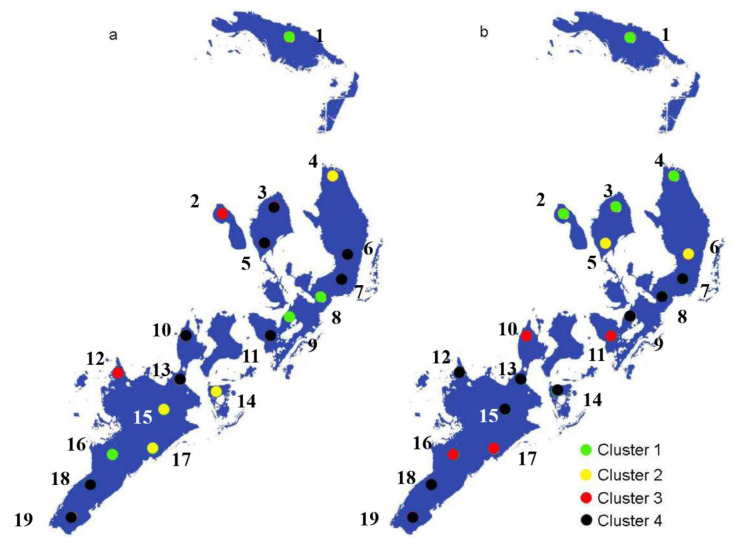
The clustering distribution of sampling points (Ward’s method) based on the HCA results between the irrigation period (**a**) and non-irrigation period (**b**).

**Figure 4 ijerph-17-05054-f004:**
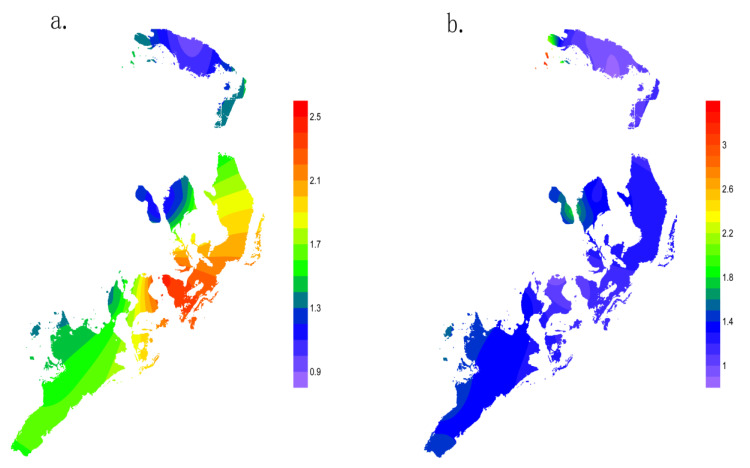
The Nemerow pollution index during the irrigation period (**a**) and non-irrigation period (**b**) stages.

**Table 1 ijerph-17-05054-t001:** Physical and chemical parameter analysis methods and detection levels.

Item *	Analysis Method	Testing Instrument	The Lowest Detection Level
pH	Glass electrode method	pH meter	0.1
NH_3_–N	Nessler’s reagent spectrophotometry	SK-100AR Ammonia nitrogen analyzer	0.025 mg/L
DO	Iodine quantity method	Laboratory glassware for titration	0.2mg/L
BOD	Dilution and inoculation method	Biochemical incubator	2 mg/L
Turbidity	Turbidity meter method	Portable turbidimeter	
Salinity	Weight method	Electronic balance	2 mg/L
Transparency	Plug’s plate method	Plug’s plate	10mm
Chlorophyll a	Acetone extraction—spectrophotometric method	Spectrophotometer	0.04mg/L
Anionic surfactant	The methylene blue spectrophotometric method	Spectrophotometer	0.05 mg/L
Suspended matter	Weight method	Electronic balance	4 mg/L
Cyanide	The isonicotinic acid-barbituric acid spectrophotometry	Flow injection analyzer (FIA)	0.001 mg/L
TN	Peroxide potassium sulfate-ultraviolet spectrophotometry	Spectrophotometer	0.05 mg/L
TP	Mo-Sb anti-spectrophotometer	Spectrophotometer	0.01 mg/L
KMnO_4_	Acid electric process	Laboratory glassware for titration	0.5 mg/L
Petroleum	Infrared spectrophotometry	Infrared oil content analyzer	0.018 mg/L
Volatile phenol	4-aminoantipyrene spectrophotometric method	Flow injection analysis (FIA)	0.001 mg/L
Sulfide	The amino dimethyl aniline photometric method		0.02 mg/L
Fluoride	Ion selective electrode potentiometry	Fluoride ion selective electrode	0.05 mg/L
Cr^6+^	1,5-diphenylcarbazide spectrophotometry	Spectrophotometer	0.004 mg/L
COD	Potassium dichromate method	Laboratory glassware for titration	30 mg/L
Se	Atomic fluorescence spectrometry	Atomic Fluorescence Spectrometer (AFS) - 830	0.002 mg/L
Zn	Flame atomic absorption spectrophotometry		0.005 mg/L
Cu	Graphite furnace atomic absorption spectrometry	NovAA-400PGraphite furnace	0.01 mg/L
Pb			0.001 mg/L
Cd			0.0001 mg/L
Hg	Atomic fluorescence spectrophotometry	Atomic Fluorescence Spectrometer (AFS) -830	6.00 × 10^−6^ mg/L
As			6.00 × 10^−5^ mg/L
Coliform bacteria	Multi-tube zymolytic method	Incubator	10 most probable number/L

***** BOD means biochemical oxygen demand; DO is dissolved oxygen; TN is total nitrogen; TP is total phosphorus; KMnO_4_ means potassium permanganate index, and COD is chemical oxygen demand.

**Table 2 ijerph-17-05054-t002:** Grading standard for water quality classification.

Grade	Comprehensive Pollution Index (*P*_total_)	Level
I	≤0.20	Cleanness
II	0.21–0.40	Sub-cleanness
III	0.41–1.00	Slight pollution
IV	1.01–2.0	Moderate pollution
V	≥2.01	Severe pollution

**Table 3 ijerph-17-05054-t003:** Descriptive statistics of water quality factors in Wuliangsuhai Lake.

Item	Range	Min	Max	Mean	Median	Standard Deviation	Variation Coefficient	Nation Standard
pH	1.42	7.89	9.31	8.43	8.41	0.27	0.03	6–9
Turbidity	64.00	3.00	67.00	14.11	11.50	10.21	0.72	≤19
Total Suspended solids(TSS)	97.00	4.00	101.00	18.72	14.00	15.51	0.83	None
Salinity	5128.00	696.00	5824.00	1902.25	1849.00	853.32	0.45	None
Transparency	180.00	10.00	190.00	87.18	90.00	41.73	0.48	None
Chlorophyll a	0.12	0.01	0.12	0.02	0.01	0.01	0.94	≤0.01
DO	6.10	2.90	9.00	5.74	5.81	1.36	0.24	≥5
KMnO_4_	8.50	3.80	12.30	8.10	7.93	2.01	0.25	≤6
BOD	9.50	2.00	11.50	3.40	2.95	1.48	0.43	≤4
COD_Mn_	97.00	16.00	113.00	41.12	38.7	16.21	0.39	≤20
TN	4.57	0.97	5.54	1.72	1.54	0.78	0.45	≤1
NH_3_–N	1.64	0.03	1.67	0.19	0.15	0.20	1.05	≤1
TP	0.2050	0.0170	0.2220	0.0792	0.0076	0.0385	0.49	≤0.2
Oil	0.0490	Ld *	0.0490	0.0037	Ld *	0.0100	2.70	≤0.05
Fluoride	0.7800	0.3100	1.0900	0.5746	0.5505	0.1331	0.23	≤1.0
Anionic Surfactants	0.1550	Ld *	0.1550	0.0189	Ld *	0.0391	2.06	≤0.2
As	0.0086	0.0008	0.0093	0.0025	0.0022	0.0015	0.58	≤0.05
Hg	0.0001	Ld *	0.0001	0.00001	0.00003	0.00001	0.40	≤0.0001
Pb	0.0436	Ld *	0.0436	0.0038	0.0021	0.0054	1.43	≤0.05
Cu	0.0216	Ld *	0.0216	0.0037	0.0020	0.0044	1.20	≤1.0
Zn	0.3480	Ld *	0.3480	0.0331	0.0180	0.0528	1.60	≤1.0
Cd	0.0010	Ld *	0.0010	0.0002	0.0002	0.0002	0.87	≤0.050
Se	0.0030	Ld *	0.0030	0.0002	0.0001	0.0004	1.86	≤0.01
Coliform Bacteria	328.00	2.00	330.00	46.56	20.00	67.88	1.46	≤1000

* Limit of detection.

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
