# Peer review of "Comprehensive Assessment of Water Quality and Pollution Source Apportionment in Wuliangsuhai Lake, Inner Mongolia, China"

_ijerph, 2020, doi:10.3390/ijerph17145054_

Round 1

Reviewer 1 Report

This study investigated spatial variation of water quality in irrigation period and non-irrigation period of 2015 in the Wuliangsuhai Lake by applying the multivariate statistical techniques and discussed the potential pollution source. The findings from this study could provide some guidance for water pollution controls of the study area. However, the author need to address the following comments before the paper can be accepted for publication.

Major comments:

1) The English language need to be improved. Please ask for professional English editing service to improve the English language.

2) In the introduction, it is suggested to provide the previous studies in Wuliangsuhai Lake and discuss the similarity and difference between this study and previous studies. And the purpose and significance of this study should be clarified clearly.

4) In the materials and methods, there were many parameters in equations for calculating comprehensive pollution index without clear definition, as Fi, ri…. How you determining the grading standards given in table 2 was unclear.

5) In the results and discussion section, please explain the conclusion “thus, industrial and domestic sewage in the Yellow River were determined to be major pollution sources.” while just by the description statistics of major heavy metals and anionic surfactants. In section 3.1, pollution source had been determined, then in section 3.4, possible source of pollutants was discussed again. Please clarify the difference the pollution source in section 3.1 and 3.4.

6) The description about nonparametric tests method should be put in Section 2.

7) The authors considered the eight components in the irrigation period and six components in non-irrigation period. Please provide the related reference to support it. In data interpretation, PCA was generally applied to reduce the parameter dimensions of the original dataset and generate a new set of linearly uncorrelated principal components with carrying most information about original data. In this study, author only selected the water quality factors of highest weighted value from each component and named them as “principle components” (line 179, line 181). Does that contradict with the expression from line 98 to line 101? Please explain it. And those factors with same or larger value was discarded. As in irrigation period, the result of PCA ingredient matrix indicated that BOD (0.756) was almost equal to COD (0.761) in principal component 1; absolute value of Coliform bacteria (-0.596) was greater than Hg (0.489) in principal component 4…. Please explain why only choose the highest one in each component and clarify the relationship between components and water quality factors in the PCA ingredient matrix (table 7).

8) In the results and discussion, the related reference about discussion of possible pollution source and the comparation with previous studies should be added.

Minor comments:

Line 19: please revise “hierarchical cluster analysis (CA)” as “hierarchical cluster analysis (HCA)”.

Please change word “nonirrigation” to “non-irrigation” in paper.

Line 19: “hierarchical cluster analysis (CA)”, line 110 “Hierarchical agglomerative clustering”, line 114 “Hierarchical agglomerative CA”, please revise them in same way.

Line 91: please explain how the PCA summarize the correlation among water quality parameters.

Line 118: please cite the related reference of comprehensive pollution index method.

Line 120 to line 123, line 129: please label each equation with a unique number.

Line 125: please check the sentence “the measured concentration of the pollutant (mg/L, pH: Dimensionless, Coliform bacteria: Ind)”, is it pollutant concentration or water quality factors?

Line 137: the maximum of pH was 9.31, which is exceeded the range of 6-9. Please revise the sentence.

Line 155: in table 4, the last column was unnamed. Please add its name.

Author Response

  • This study investigated spatial variation of water quality in irrigation period and non-irrigation period of 2015 in the Wuliangsuhai Lake by applying the multivariate statistical techniques and discussed the potential pollution source. The findings from this study could provide some guidance for water pollution controls of the study area. However, the author need to address the following comments before the paper can be accepted for publication. Major comments: The English language need to be improved. Please ask for professional English editing service to improve the English language.

Response: Thank you. The manuscript is revised according to all your suggestions. The English language was revised through AJE (https://www.aje.com/) before the last submission.

  • In the introduction, it is suggested to provide the previous studies in Wuliangsuhai Lake and discuss the similarity and difference between this study and previous studies.

Response: Thank you for your suggestion. We add more than ten articles to describe the previous studies in Wuliangsuhai Lake. At the same time, according to the summary of literatures, we highlight the differences between this study and previous works.

  • And the purpose and significance of this study should be clarified clearly.

Response: Thank you for your suggestion. We rewrite the last paragraph to clarify the purpose more clearly

  • In the materials and methods, there were many parameters in equations for calculating comprehensive pollution index without clear definition, as Fi, ri…. How you determining the grading standards given in table 2 was unclear.

Response: Thank you for your suggestion. All the definitions of parameters in equations were checked again, including Fi, Ci, Si, Smax, Fmax , Fw, , ri, and n. We reorganize this paragraph and briefly describe the calculation steps as follows: “Nemerow pollution exponential method is one of the most commonly used comprehensive assessment approaches, which allows the assessment of the overall degree of water pollution and includes the contents of all analyzed water-quality factors. The factor with the largest weight (FwMax) among all water-quality factors is determined according to Eq. (2-1) and Eq. (2-2). The Ci/Si ratio of the FwMax is considered in Eq. (2-3). Then, the improved Nemerow comprehensive pollution index was calculated as shown in Eq. (2-3) and Eq. (2-4). The average values determined for  were conducted to assess the water quality of the lake (Table 2).”

  • In the results and discussion section, please explain the conclusion “thus, industrial and domestic sewage in the Yellow River were determined to be major pollution sources.” while just by the description statistics of major heavy metals and anionic surfactants. In section 3.1, pollution source had been determined, then in section 3.4, possible source of pollutants was discussed again. Please clarify the difference the pollution source in section 3.1 and 3.4.

Response: Thank you for your suggestion. Here we delete the sentence “thus, industrial and domestic sewage in the Yellow River were determined to be major pollution sources.”, and that was discussed in section 3.4 according to statistics, PCA, and CA results.

  • The description about nonparametric tests method should be put in Section 2.

Response: Thank you for your suggestion. We move the description of the nonparametric tests method to section 2.

  • The authors considered the eight components in the irrigation period and six components in non-irrigation period. Please provide the related reference to support it. In data interpretation, PCA was generally applied to reduce the parameter dimensions of the original dataset and generate a new set of linearly uncorrelated principal components with carrying most information about original data. In this study, author only selected the water quality factors of highest weighted value from each component and named them as “principle components” (line 179, line 181). Does that contradict with the expression from line 98 to line 101? Please explain it. And those factors with same or larger value was discarded. As in irrigation period, the result of PCA ingredient matrix indicated that BOD (0.756) was almost equal to COD (0.761) in principal component 1; absolute value of Coliform bacteria (-0.596) was greater than Hg (0.489) in principal component 4…. Please explain why only choose the highest one in each component and clarify the relationship between components and water quality factors in the PCA ingredient matrix (table 7).

Response: Thanks for your suggestions. The whole data used for PCA analysis, the loading plots, and factor score results were provided. The discussion was revised according to the results. The tables of PCA analysis results were moved into supplementary information. Eight components of the PCA analysis explained 69.404% of the total variance. Each component was characterized by the water-quality parameters with higher-weighted values (Table S3). Thus, the eight principle components were dominated with BOD, CODMn, and salinity (PC1); turbidity, TSS, Transparency, and TP (PC2); pH, Zn, and Cd (PC3); As and Hg (PC4); KMnO4 (PC5); TN and NH3-N (PC6); fluoride, Pb, Cu, and Se (PC7); chlorophyll a (PC8). Different contributions of PCA components during non-irrigation and irrigation periods were observed in Figure 2. The measured data of irrigation periods were characterized by PC8, PC6, PC3, while that of non-irrigation was characterized by PC2, PC1, PC4, and PC7. Thus, nutrient parameters, heavy metal parameters (Zn and Cd), and pH reflected the degree of pollution of the lake during the irrigation period, suggesting that eutrophication is a significant problem during this stage.

  • In the results and discussion, the related reference about discussion of possible pollution source and the comparation with previous studies should be added.

Response: Thanks for your suggestions. We add the related references to discuss the possible pollution sources, and also we compared our results with the previous studies.

Minor comments:

  • Line 19: please revise “hierarchical cluster analysis (CA)” as “hierarchical cluster analysis (HCA)”.Please change word “nonirrigation” to “non-irrigation” in paper. Line 19: “hierarchical cluster analysis (CA)”, line 110 “Hierarchical agglomerative clustering”, line 114 “Hierarchical agglomerative CA”, and please revise them in same way.

Response: Thank you. We revise them according to your suggestions.

  • Line 91: please explain how the PCA summarize the correlation among water quality parameters.

Response: Thank you for your suggestion. We revise this sentence. PCA was applied for the assessment of the irrigation effects on water quality, and each water-quality parameter was standardized before PCA to minimize the influence of different variables and their respective units of measurements. Besides, in the PCA section, we described the process of PCA analysis.

  • Line 118: please cite the related reference of comprehensive pollution index method.

Response: Thank you for your suggestion. The reference was added.

  • Line 120 to line 123, line 129: please label each equation with a unique number.

Response: Thank you for your suggestion. We label each equation with a unique number.

  • Line 125: please check the sentence “the measured concentration of the pollutant (mg/L, pH: Dimensionless, Coliform bacteria: Ind)”, is it pollutant concentration or water quality factors?

Response: Thank you for your suggestion. We change it into water quality factors.

  • Line 137: the maximum of pH was 9.31, which is exceeded the range of 6-9. Please revise the sentence.

Response: Thank you for your suggestion. We change it into water quality factors.

  • Line 155: in table 4, the last column was unnamed. Please add its name.

Response: Thank you for your suggestion. The median of values is added in table 4, and also we delete the last column.

Reviewer 2 Report

I think the idea interesting and valuable, although some minor concerns arise when reading the document: 1. the contributions are not well elaborated in Introduction section, which is important to distinguish your paper with others. 2. lack of articles published in the near 3 years. 3. titles in section 3.2 and section 3.3 are not suitable, please change them, 4. managerial insights should be added in the discussion, in this section, you can tell the reader what problems you have solved and what solutions you have offered to them. In my opinion, it is vital for a well-qualified paper. 5. future research direction should be added, in addition to the content of success risk in the future

Author Response

(1) I think the idea interesting and valuable, although some minor concerns arise when reading the document: the contributions are not well elaborated in Introduction section, which is important to distinguish your paper with others.

Response: Thank you for your suggestion. We revise the introduction and describe the previous studies in Wuliangsuhai Lake. At the same time, according to the summary of literatures, we highlight the differences between this study and previous works.

 (2)Lack of articles published in the near 3 years.

Response: Thank you for your suggestion. We add the recent articles in recent years related to Wuliangsuhai Lake.

(3)Titles in section 3.2 and section 3.3 are not suitable, please change them,

Response: Thank you for your suggestion. The titles in section 3.2 and section 3.3 are revised.

(4)Managerial insights should be added in the discussion, in this section, you can tell the reader what problems you have solved and what solutions you have offered to them. In my opinion, it is vital for a well-qualified paper.

Response: Thank you for your suggestion. In the Introduction and Conclusion sections, we added a description of the highlighted research in this article. Besides, the related references about the discussion of possible pollution sources, and the comparison with previous studies are added in the results and discussion sections.

(5)Future research direction should be added, in addition to the content of success risk in the future

Response: Thank you for your suggestion. We rewrite the conclusions, and the future research direction is added.

Reviewer 3 Report

The article has serious flaws and cannot be published in the present form. In my opinion, the authors used statistical and chemometric techniques without having the competencies to do so. This led to a presentation and an interpretation of results which is substantially wrong. I suggest contacting a chemometrician in order to seriously improve the Results and discussion section.

Specific comments:

Line 17-18: This sentence has no meaning and simply repeats what stated in the previous and following sentences. I suggest removing it.

Lines 21-24: Talking about specific clusters in the abstract is pointless. The reader has no idea of which samples or analytes are included in those clusters. Please modify this paragraph in order to explain the main results without talking about clusters or to specify what these clusters are composed of.

Lines 68-69: Did the authors collect sample duplicates or for each sampling site and each sampling time only one sample was collected?

Lines 83-86: What do the authors mean with “official”? If the detection limits were not experimentally determined, a reference must be provided. Moreover, “6+” must be superscript.

Tables 1 and 4, lines 144 and 158: “3” in NH3-N must be subscript

Line 92: Which standardization method was used? The most common method is autoscaling, but many different methods can be used. Moreover, the authors did not explain the approach they used with results below the detection limit.

Line 99: Technically, this is not true. PCA converts intercorrelated variables into an EQUAL NUMBER of independent variables. Then, considering that the great majority of the variance of the dataset is retained by the first few variables, only the first few PCs are taken into account for the dataset interpretation.

Lines 113-114: Even though I understand what the authors mean with “a dramatic reduction in the dimensionality of the original data”, I believe that this expression it is formally wrong. I suggest rephrasing it.

Line 115: Squared Euclidean distances are used as a measure of dissimilarity, not of similarity.

Tables 2 and 3: “i” and “total” in Pi and Ptotal should be subscript

Line 140: As is not a metal

Lines 141 and 158: “6+” must be superscript

Lines 140-143: In the same sentence, the authors state that Hg and Pb were below the detection limit and close to the permissible limit of SEPA. Please re-phrase it.

Table 4: For environmental data, the median is more appropriate than the mean. Please add the median to the table. Moreover, the heading of the last column is missing, and the acronym “Ld” must be explained (I suggest adding a table footnote). I also suggest including in the Supplementary Material a table providing all the sample results.

Lines 157-161: I do not understand what variation > or = 1 means. Variation from what to what? And why 1? Is there a ratio of something?

Lines 162-164: What was the Kruskal-Wallis test used for? What is the purpose of the test, in this context? I suppose the authors wanted to verify if the difference between the samples collected in the various sampling points was significant, but it should be clearly stated when the test is presented (and not only where the results are interpreted).

Table 5: The table itself says nothing. I suggest the authors to make a pairwise comparison (such as Conover-Iman) in order to be able, for each variable, to group samples according to their similarity. The authors should also clearly report how they decided to group the samples (How many groups? Which criteria was chosen for grouping the samples?). The Kruskal-Wallis test only says if there is a significant difference between one group and the others, but it does not say anything else: neither which group is different nor if more than one group are different. When the Kruskal-Wallis test says that at least one group is different, a pairwise comparison is necessary for interpreting the results.

Lines 176-177: Generally, only the first 2-3 (or, in few cases, 4) PCs are worth interpretation. The other PCs only represent environmental and analytical noise. The scree plot is commonly used for deciding how many variables should be considered.

Lines 177-182: This is wrong! No variables can be excluded from PCA!

Lines 186-188: The authors state “numerous studies” but only one reference is provided.

Tables 6 and 7: These tables are not the results of PCA and mean nothing. When PCA is performed, the score and loading plot (of the whole dataset!!) must be provided, and they must be jointly interpreted.

Lines 205-212: The authors simply described the figure, without giving any interpretation of the results. Moreover, they describe the clusters by using the site numbers, but they never showed these numbers on a map.

Figure 2: The figure is nice, but it is NOT a dendrogram. I suggest reporting both this figure and a CA dendrogram. Moreover, the caption states that this figure was obtained by PCA scores, yet it was obtained by CA.

Lines 226-231: Are these authors’ hypothesis? It should be stated, since the authors cannot be sure of their interpretation. How they reached these conclusions? Are there any references, any previous studies stating the same?

Lines 242-246: This is not true. For achieving this objective, discriminant analysis methods and a huge set of measurements are needed. Discriminant analysis methods imply the use of, besides a training set, a randomly-chosen evaluation set, which is used for calculating the forecasting ability of the model.

Author Response

The article has serious flaws and cannot be published in the present form. In my opinion, the authors used statistical and chemometric techniques without having the competencies to do so. This led to a presentation and an interpretation of results which is substantially wrong. I suggest contacting a chemometrician in order to seriously improve the Results and discussion section.

Response: Thank you for your suggestion. We carefully made a large number of modifications in accordance with your comments, especially for the problems of statistical methodology.

Specific comments:

  • Line 17-18: This sentence has no meaning and simply repeats what stated in the previous and following sentences. I suggest removing it.

Response: Thanks. We have removed this sentence.

  • Lines 21-24: Talking about specific clusters in the abstract is pointless. The reader has no idea of which samples or analytes are included in those clusters. Please modify this paragraph in order to explain the main results without talking about clusters or to specify what these clusters are composed of.

Response: Thanks. We modify this paragraph. The results showed that, during the irrigation period, sites in the middle part of the lake (clusters 1 and 3) had a high pollution level due to receiving most of the agricultural and some industrial wastewater from the Hetao irrigation area. During the non-irrigation period, the distribution of the comprehensive pollution index was the reverse to that seen during the irrigation period, and the degree of pollutant index was obviously reduced.

  • Lines 68-69: Did the authors collect sample duplicates or for each sampling site and each sampling time only one sample was collected?

Response: Thanks. Each water sample is mixed water columns at different depths. We modify the statement,“Water samples were collected from 19 sites (Figure 1b) at bimonthly intervals between June 2015 and November 2015 in Wuliangsuhai Lake. Each sample was composed of three mixed water columns at depths of 0.5 m, 1 m, and 2 m.”

  • Lines 83-86: What do the authors mean with “official”? If the detection limits were not experimentally determined, a reference must be provided. Moreover, “6+” must be superscript.

Response: Sorry that we misused the description. The concentrations were below the “method” detection limit. This sentence is edited as “A total of 28 hydrochemical variables were analyzed; however, the concentrations of volatile phenols, cyanide, sulfide, and Cr6+ for all samples were below detection levels (0.001 mg/L for volatile phenol and cyanide, 0.02 mg/L for sulfide, and 0.004 mg/L for Cr6+).” And also, “Cr6+” is changed into “Cr6+”.

  • Tables 1 and 4, lines 144 and 158: “3” in NH3-N must be subscript

Response: Thanks. We edit all the subscripts.

Line 92: Which standardization method was used? The most common method is autoscaling, but many different methods can be used. Moreover, the authors did not explain the approach they used with results below the detection limit.

Response: Thanks for your suggestions. Each water-quality parameter was standardized by using euqation . All the chemical parameters analyses including some results below the detection limit were carried out according to the Chinese national surface water and wastewater monitoring technology standards. And also, the methods were listed in table 1.

  • Line 99: Technically, this is not true. PCA converts intercorrelated variables into an EQUAL NUMBER of independent variables. Then, considering that the great majority of the variance of the dataset is retained by the first few variables, only the first few PCs are taken into account for the dataset interpretation.

Response: We fully agree with your suggestion. The description for PCA has been modified according to your positive suggestion as follows: “PCA is a method of mathematical transformation that attempts to reduce the dimensionality of datasets. The set of relevant variables is converted into another equal number of independent variables that is not related by a linear transformation, and these new variables are then arranged in descending order of variance”.

  • Lines 113-114: Even though I understand what the authors mean with “a dramatic reduction in the dimensionality of the original data”, I believe that this expression it is formally wrong. I suggest rephrasing it.

Response: Thanks. We rephrase this sentence. “The dendrogram provides a summary of clustering processes, presenting a visual picture of the groups and their proximity.”

  • Line 115: Squared Euclidean distances are used as a measure of dissimilarity, not of similarity. Tables 2 and 3: “i” and “total” in Pi and Ptotal should be subscript. Line 140: As is not a metal. Lines 141 and 158: “6+” must be superscript

Response: Thanks. The formats and description problems have been revised.

  • Lines 140-143: In the same sentence, the authors state that Hg and Pb were below the detection limit and close to the permissible limit of SEPA. Please re-phrase it.

Response: Thanks. We revise this sentence.

  • Table 4: For environmental data, the median is more appropriate than the mean. Please add the median to the table. Moreover, the heading of the last column is missing, and the acronym “Ld” must be explained (I suggest adding a table footnote). I also suggest including in the Supplementary Material a table providing all the sample results.

Response: Thanks. The median is added in table 4, and also we delete the last column. LD means the limit of detection that is added in the table footnote. All the sample results were supplied in the Supplementary Material

  • Lines 157-161: I do not understand what variation > or = 1 means. Variation from what to what? And why 1? Is there a ratio of something?

Response: We rephrase this sentence: the coefficients of variation for NH3-N, oil, anionic surfactants, Se, Zn, coliform bacteria, Pb, Cu, chlorophyll, Cd and TSS values were relatively high, indicating these water-quality factors were significantly different in terms of their temporal and spatial distribution.

  • Lines 162-164: What was the Kruskal-Wallis test used for? What is the purpose of the test, in this context? I suppose the authors wanted to verify if the difference between the samples collected in the various sampling points was significant, but it should be clearly stated when the test is presented (and not only where the results are interpreted). Table 5: The table itself says nothing. I suggest the authors to make a pairwise comparison (such as Conover-Iman) in order to be able, for each variable, to group samples according to their similarity. The authors should also clearly report how they decided to group the samples (How many groups? Which criteria was chosen for grouping the samples?). The Kruskal-Wallis test only says if there is a significant difference between one group and the others, but it does not say anything else: neither which group is different nor if more than one group are different. When the Kruskal-Wallis test says that at least one group is different, a pairwise comparison is necessary for interpreting the results.

Response: Thanks for your suggestions. We used the Kruskal-Wallis test to just verify if there are differences between sampling sites or periods, which suggest that parameters including TSS, salinity, transparency, KMnO4, BOD, fluoride and coliform bacteria showed significant differences among different sites, and parameters including turbidity, salinity, transparency, chlorophyll a, DO, KMnO4, BOD, CODMn, Fluoride, As, Hg, Pb, Cu, Zn, Cd, and Se showed significant differences along the sampling periods. We have revised these statements to be clearer. Table 5 is also moved to the supplementary materials.

  • Lines 176-177: Generally, only the first 2-3 (or, in few cases, 4) PCs are worth interpretation. The other PCs only represent environmental and analytical noise. The scree plot is commonly used for deciding how many variables should be considered.

Response: Thanks for your suggestion. Eight components of the PCA analysis explained 69.404% of the total variance, and have similar interpretations of the total variance. The results are all provided in supplementary materials. The important PCs are discussed in the article.

  • Lines 177-182: This is wrong! No variables can be excluded from PCA!

Response: Thanks for your suggestion. No variables are excluded and each component was characterized by the water-quality parameters with higher-weighted values (Table S3). Thus, the eight principle components were dominated with BOD, CODMn, and salinity (PC1); turbidity, TSS, Transparency, and TP (PC2); we have revised these sentences to be clearer.

  • Lines 186-188: The authors state “numerous studies” but only one reference is provided.

Response: Thanks for your suggestion. We add references.

  • Tables 6 and 7: These tables are not the results of PCA and mean nothing. When PCA is performed, the score and loading plot (of the whole dataset!!) must be provided, and they must be jointly interpreted.

Response: Thanks for your suggestions. The whole data used for PCA analysis, the loading plots and factor score results were provided, and discussion was revised according to the results. The tables of PCA analysis results were moved into supplementary information.

  • Lines 205-212: The authors simply described the figure, without giving any interpretation of the results. Moreover, they describe the clusters by using the site numbers, but they never showed these numbers on a map.

Response: Thanks for your suggestion. We describe the figure 3 with giving the interpretation of the results, and also site numbers are given in figure 3.

  • Figure 2: The figure is nice, but it is NOT a dendrogram. I suggest reporting both this figure and a CA dendrogram. Moreover, the caption states that this figure was obtained by PCA scores, yet it was obtained by CA.

Response: Thanks for your suggestion. Two CA dendrograms are added in the supporting materials. The clustering distribution of sampling points was obtained by CA results.

  • Lines 226-231: Are these authors’ hypothesis? It should be stated, since the authors cannot be sure of their interpretation. How they reached these conclusions? Are there any references, any previous studies stating the same?

Response: Thanks for your suggestion. This statement was revised. The water quality of sampling sites located in the center of the lake was likely influenced by water inflow from the inlet of the Hetao irrigation area and by the environment of the lake sediment [18, 25]. And also, previous studies have similar results.

  • Lines 242-246: This is not true. For achieving this objective, discriminant analysis methods and a huge set of measurements are needed. Discriminant analysis methods imply the use of, besides a training set, a randomly-chosen evaluation set, which is used for calculating the forecasting ability of the model.

Response: Thanks for your suggestion. We delete the incorrect statement and revise this sentence.

Round 2

Reviewer 3 Report

I noticed and appreciated that many changes were made according to my previous suggestions. However, few other changes are needed. Moreover, the track-change instrument was not properly used, since it is not possible to see which words were deleted and substituted.

Lines 79-81: Typically, chemical analysis should be performed in triplicate or, at least, in duplicate. This ensures that no casual errors were performed during the analytical procedure and increases the reliability of the results.

Line 106: The standardization method here described is called “autoscaling”. I suggest substituting the equation with this expression.

Lines 181-188: When the Kruskal-Wallis test is performed and reveals the presence of significant differences, a pairwise comparison test (such as Conover-Iman) is necessary for understanding what those differences are caused by and to properly discuss the results.

Paragraph 3.2: In principal component analysis, only the first 3 (or, in few cases, 4) principal components worth interpretation. The subsequent PCs only represent environmental and analytical noise and MUST NOT be included in the discussion, since they have NO MEANING. Obviously, the inclusion of each PC determines an increment of the percentage of the total variance explained by PCA. However, this is not a valid reason for including a huge number of PCs.
